# TRPV1 Channel: A Noxious Signal Transducer That Affects Mitochondrial Function

**DOI:** 10.3390/ijms21238882

**Published:** 2020-11-24

**Authors:** Rebeca Juárez-Contreras, Karina Angélica Méndez-Reséndiz, Tamara Rosenbaum, Ricardo González-Ramírez, Sara Luz Morales-Lázaro

**Affiliations:** 1Department of Cognitive Neuroscience, Neurosciences Division, Institute of Cellular Physiology, National Autonomous University of Mexico, UNAM, Mexico City 04510, Mexico; rjuarez@ifc.unam.mx (R.J.-C.); amendez@ifc.unam.mx (K.A.M.-R.); trosenba@ifc.unam.mx (T.R.); 2Department of Molecular Biology and Histocompatibility, “Dr. Manuel Gea González” General Hospital, Mexico City 14080, Mexico; ricardo.gonzalezr@salud.gob.mx

**Keywords:** TRPV1, nociception, mitochondrial dysfunction, ROS, apoptosis, pain

## Abstract

The Transient Receptor Vanilloid 1 (TRPV1) or capsaicin receptor is a nonselective cation channel, which is abundantly expressed in nociceptors. This channel is an important transducer of several noxious stimuli, having a pivotal role in pain development. Several TRPV1 studies have focused on understanding its structure and function, as well as on the identification of compounds that regulate its activity. The intracellular roles of these channels have also been explored, highlighting TRPV1′s actions in the homeostasis of Ca^2+^ in organelles such as the mitochondria. These studies have evidenced how the activation of TRPV1 affects mitochondrial functions and how this organelle can regulate TRPV1-mediated nociception. The close relationship between this channel and mitochondria has been determined in neuronal and non-neuronal cells, demonstrating that TRPV1 activation strongly impacts on cell physiology. This review focuses on describing experimental evidence showing that TRPV1 influences mitochondrial function.

## 1. Introduction

The interpretation of the cell’s environmental signals depends upon transducing and encoding these stimuli into a comprehensible cellular language. This selective process is facilitated by several ion channels located in the cell’s plasma membrane, among which the family of transient receptor potential (TRP) cation channels plays an essential role.

The first evidence for the existence of these channels was found in the *Drosophila melanogaster* fly. Cosens and Manning [1] analyzed a spontaneous mutant that displayed transient rather than sustained responses in the presence of prolonged bright illumination in electroretinogram (ERG) measurements [1]. This mutation rendered the flies blind and the investigators who isolated this mutant named it the “A-type” mutant, ascribing its phenotype to a failure of photopigment regeneration [2]. Nonetheless, several concerns were raised at the time of its isolation, with the main one being that the results were based on a single spontaneously occurring mutant, for which there was no description of its genetic background. Hence, there was a lack of knowledge on what genetic alterations this strain embodied with the results possibly being due to the additive effects of alterations in several genes mapping to the same chromosome [3]. Pak and collaborators proved that the observed phenotype was indeed due to mutation in a single gene by isolating multiple mutated alleles from a baseline stock of known genetic background [3]. It was also important to establish the cellular origins of the measured ERG components because it had not been determined whether the lack of a sustained response seen in the ERG of this strain originated from the photoreceptors or from other retinal cells [3]. In 1975, Minke and collaborators performed intracellular recordings from the mutant photoreceptors and were able to determine that the defect arose from the photoreceptors. Thus, they concluded that this mutant is defective in phototransduction [4]. Several other studies were performed with this mutant, which helped to forge a representative name for this mutant: the “transient receptor potential” or TRP [4,5,6].

Ten years later, the *trp* gene was isolated [7] and characterized as a genomic region that encodes for some RNA species missing in the mutant fly [8]. Interestingly, transformation of the mutant germline with the region encoding for the *trp* gene reestablished the normal phototransduction phenotype in the *trp* mutant fly, confirming the identity of this gene [7]. Finally, the cloning of the *trp* gene suggested a transmembrane protein with eight α-helices containing highly hydrophilic N- and C-termini, which was consistent with already known receptor/transporter/channel proteins [9,10].

The TRP protein was localized to the rhabdomere, and hence thought to participate in phototransduction. Nonetheless, the absence of homologous proteins to the TRP in available databases and other results suggested that in null *trp* alleles, there was still a persistent sustained receptor potential under dim light stimulation [4,11] and so it was concluded that the *trp* gene does not encode for the light-sensitive channel [9,10].

Later, it was shown that La^3+^ could mimic the *trp* phenotype [12], and this led to the proposal that the *trp* gene encoded for an inositide-activated Ca^2+^ channel/transporter required for Ca^2+^ store refilling [13]. Voltage-clamp recording in the whole-cell configuration showed that the primary defect in the *trp* mutant was a drastic reduction in the Ca^2+^ permeability of the light-sensitive channels themselves [14].

Then, another protein similar to the *trp* gene product was found, the TRP-like (TRPL) [15]. The product encoded by the *trpl* gene exhibited characteristics of a transmembrane protein that contains two putative calmodulin-binding sites and an ankyrin-like repeat domain [15]. Sequence homology analyses demonstrated that the central transmembrane region of the *trp* and *trpl* products resemble the transmembrane segments (S1–S6) of calcium channels, suggesting that these TRP proteins could act as ion channels [15]. The functional description of invertebrate TRP proteins demonstrated that these proteins function as light-sensitive ion channels that allow calcium entry in *Drosophila* photoreceptors [14], and the phenotype of the *trp* mutation was reinterpreted and it was suggested that the light response of *Drosophila* is mediated by channels composed from the *trp* and *trpl* gene products [14,15]. In summary, there are two distinct conductances that give rise to the normal light-sensitive current: one encoded by the *trp* gene that is highly Ca^2+^ selective, and a second channel that is responsible for the residual light-sensitive current in the *trp* mutants encoded by the homologous gene *trpl*.

Remarkably, the *trp/trpl* gene products lack the presence of several positively-charged amino acids in their S4, the region that typically constitutes the voltage sensor in Ca^2+^ channels, suggesting that these invertebrate TRP proteins lack this domain. These features were the first clues about the structure of invertebrate proteins belonging to the TRP family of ion channels.

Later, these structural characteristics were identified in the first mammalian homolog of the *TRP* gene isolated from human fetal brain cDNA libraries. It was named the transient receptor potential channel-related 1 (TRPC1), a protein with similar structural organization to that of TRP proteins from invertebrates [16]. This, in turn, led to the isolation and characterization of several TRP channels, resulting in discoveries pertaining to their pivotal roles in mammalian physiology.

To date, TRP channels represent an extended and essential family of ion channels composed of 28 members divided according to homology in their primary sequences. This family is classified into seven subfamilies: TRPC (canonical), TRPV (vanilloid), TRPM (melastatin), TRPP (polycystin), TRPML (mucolipin), TRPA (ankyrin) and TRPN (no mechanoreceptor), and the latter is exclusively found in invertebrates and fish [17] (Figure 1). Additionally, another subfamily called TRPY (yeast) has been identified in vacuoles from *Saccharomyces cerevisiae* [18], although the details of its physiological roles remain unknown.

Interestingly, TRP channel localization and function are not limited to the cell surface. Some of these channels are intracellularly located in the membranes of organelles, regulating ion flux between the cytosol and these intracellular compartments [19,20]; furthermore, cell surface-located channels can also regulate calcium homeostasis in some organelles, such as the mitochondria. In this review, we will describe how TRPV1 activation modifies mitochondrial function. Here, we will discuss the data showing that the channel’s activation can induce mitochondrial dysfunction in neuronal and non-neuronal cells.

## 2. The TRPV1 Channel: A Key Transducer of Nociception

The discovery of the TRP channel, known as the “capsaicin” receptor or TRPV1, highlighted its essential role in mammalian nociception [21,22,23]. Its isolation and cloning represented a significant step in the advancement of the pain research field, providing initial evidence of the relationship between the structure, the activity, and the physiological role of this channel [21,22,23].

TRPV1 cDNA was isolated from a rodent dorsal root ganglion (DRG) library and expressed in oocytes to perform membrane recordings, confirming that capsaicin evokes currents through the activation of this channel [22]. Full characterization of the activity and properties of TRPV1 showed that it is a nonselective cation channel which exhibits preference to Ca^2+^ ions [22].

The TRPV1 protein is abundantly expressed in small and medium diameter sensory neurons from trigeminal and dorsal root ganglia (TG and DRG, respectively), as well as in the nodose ganglia [22,23,24]. Its expression is mainly associated with Substance P-producing DRG neurons, which are widely related to pain perception [23]. Additionally, low TRPV1 expression has been reported in specific regions of the central nervous system, and in non-neuronal cells [25,26,27], where elucidation of its roles in these areas has been extensively studied.

Additionally, it was demonstrated that natural compounds activate the polymodal TRPV1 channel. Examples of these compounds are capsaicin, resiniferatoxin and allicin (the pungent compound of chili peppers and chemical compounds found in *Euphorbia resinifera* and garlic, respectively) [22,28]. Also, TRPV1 is activated by noxious heat (around 42 °C) and by extracellular acid and basic intracellular pH [22,23,29]. Furthermore, TRPV1 is activated by some endogenous compounds, such as anandamide, lysophosphatidic acid (LPA), and by metabolites derived from arachidonic acid and linoleic acid byproducts, among others [30,31,32,33].

Transgenic mice lacking TRPV1 expression (TRPV1-KO) evidenced that this channel is a crucial transducer of several noxious signals that converge to integrate painful stimuli, since these mice display impaired nociception [21].

Until now, several reports support the essential role of TRPV1 modulation through modifications such as phosphorylation, which is associated with lowering the activation threshold, usually referred as sensitization [34]. In contrast, dephosphorylation or reduction of the channels available on the cell surface causes desensitization [35,36]. These phenomena are crucial in hyperalgesia and chronic pain events, where inflammatory molecules sensitize TRPV1 [37].

### 2.1. TRPV1 Structural Features

Prediction of the TRPV1 protein structure suggested it was a protein containing six transmembrane segments (S1–S6) joined by intracellular and extracellular loops [22], where the re-entrant loop between S5 and S6 form the pore or ion conduction pathway when the channel is assembled as a homotetramer (Figure 2a).

The first 3D structural details of TRPV1 were obtained using X-ray crystallographic techniques [38]. These assays confirmed the presence of a domain with six ankyrin repeats (ARD), each consisting of a pair of antiparallel α-helices linked by a finger loop. Functionally, these ankyrin repeats have an important role for the interaction between ATP and calmodulin (CaM) [38]. Additionally, a CaM binding site located in the C-terminal region of TRPV1 was also described [39,40], suggesting that CaM plays a critical role in TRPV1 modulation.

A few years later, single-particle electron cryo-microscopy (cryo-EM) allowed for the solving of the first TRPV1 three-dimensional (3D) structure at a low resolution (19 Å) [41]. This structure confirmed the tetrameric channel conformation with a large open basket-like domain corresponding to the intracellular N- and C-termini, and a compact transmembrane domain that resembles that of the transmembrane domain structure of voltage-gated potassium (Kv) and sodium (Nav) channels [41]. TRP channels in general resemble voltage-gated ion channels. They display reminiscent structural features such as the presence of an antiparallel β-sheet from the linker region and a β-strand from the C-terminus region that make contact with two of the ankyrin repeats of an adjacent subunit (which presumably is important for channel assembly) [42]. These features are similar to the T1 domain of Kv channels where this region plays an important role in subunit assembly [43]. On the other hand, the wider outer pore and the shorter selectivity filter of TRPV1 also point to similarities with bacterial Nav channels [42].

Then, a better resolution (3.4 Å) 3D structure for TRPV1 was obtained, evidencing amino acid side chains and β-sheets located in the cytosolic domain of the protein [42] (Figure 2b). This ultra-structural analysis also described the tetrameric TRPV1 structure with a central ion conduction pore conformed by two transmembrane α-helices (S5–S6) and a short pore helix (S5-P-S6), together with a voltage-sensor like domain consisting of a cluster of four transmembrane α-helices (S1–S4) [42] (Figure 2b). The structural details of TRPV1 were primarily confirmed by this new model, suggesting that the TRPV1 channel shares many structural features with voltage-gated channels and potassium channels. However, unlike Kv channels, TRPV1 is not strongly voltage-dependent due to the lack of positively-charged residues in S4 [42].

Moreover, this approach also allowed for the resolution of the “TRP domain” within the C- terminus. It was revealed that this conserved 25 amino acid domain consists of an α-helical structure parallel to the membrane, that encompasses the first two-thirds of the TRP domain and interacts with the S4–S5 linker (Figure 2b). Thus, the TRP helix seems to have a critical role in integrating allosteric modulation between the domains of the channel [42].

### 2.2. Intracellular TRPV1 Localization

After the discovery of TRPV1 as a protein expressed in the plasma membrane of small and medium-sized DRG neurons [22,23,24], its intracellular location was also reported [24,44,45,46] (Table 1). Initially, TRPV1 localization in organelles was determined in DRG neurons and TRPV1-expressing COS7 cells by using confocal and electron microscopy, demonstrating for the first time that TRPV1 was found in intracellular compartments (such as the ER) of these cells [44,46,47,48]. Further reports also revealed internal TRPV1 localization in the mitochondria and ER of TRPV1-expressing HEK293 cells, cardiomyocytes, glial cells, and muscle cells, among others [45,49,50,51,52] (Table 1). Intracellular localization in diverse types of TRPV1-expressing cells highlights its specificity and rules out effects of overexpression of the channel in transfected HEK293 cells [53] (Table 1).

TRPV1 expression has also been detected in rat skeletal muscle, where this channel is mainly located in the sarcoplasmic reticulum [60], and it has been suggested to contribute to the regulation of excitation and contraction through Ca^2+^ homeostasis [59]. Particularly, TRPV1 has been isolated from membrane fractions located at longitudinal sarcoplasmic reticula [59]. Besides, proximity ligation assays achieved in human airway smooth muscle (ASM) cells indicated that TRPV1 is expressed in a fashion where it is closely associated to sarco/endoplasmic reticulum Ca^2+^-ATPase (SERCA) pumps [50].

Furthermore, the co-localization of TRPV1 has been reported with the 58 K Golgin protein (a marker of the Golgi complex) in microglial cells, suggesting its localization in this internal cell compartment [49]. Also, TRPV1 protein has been visualized in the Golgi compartment of small-diameter DRG neurons and forming aggregates in the Golgi of some breast cancer cell lines [24,61] (Table 1). These findings could suggest the presence of an available pool of functional channels located in the Golgi or of channels being transported to the plasma membrane, an observation that needs more detailed analysis.

Remarkably, several reports support the intracellular availability of TRPV1 channels in the form of functional pools, enhancing the versatility of this channel as a cell surface signal transducer and a regulator of Ca^2+^ mobilization between the cytosol and some organelles [45,46,47]. Additionally, several reports show that the activation of TRPV1 channels located in the cell surface also play an essential role in regulating Ca^2+^ homeostasis of intracellular compartments, such as the mitochondria, as is described below.

### 2.3. TRPV1 Channels: Their Mitochondrial Actions in Non-Neuronal Cells

Mitochondrial TRPV1 localization has been mainly reported in non-neuronal cells; for example, it has been detected in cardiac mitochondrial fractions derived from mice [62]. Moreover, using confocal microscopy, it has been determined that TRPV1 displays co-localization with mitochondrial-resident proteins in primary neonatal cardiomyocytes [51]. Similarly, human parietal cells and human odontoblast-like cells exhibit positive mitochondrial immunostaining for TRPV1 [55,56], and recently, it has been reported that this channel is expressed in endothelial cells and found in the mitochondria [57] (Table 1).

Reports from different laboratories have shown that TRPV1 activation has beneficial consequences for mitochondrial function and regulates cell physiology. For example, an initial report demonstrated that microglia derived from WT mice display TRPV1 channels that are abundantly co-localized with specific markers of mitochondria, ER, Golgi complex and lysosomes. However, intriguingly, these cells lack TRPV1 at their cell surface [49]. Accordingly, whole-cell patch-clamp recordings showed that microglia lack capsaicin-evoked currents, although they display increased intramitochondrial Ca^2+^ produced by TRPV1 activation that lead to mitochondrial depolarization. These observations were obtained using time-lapse imaging of an intramitochondrial Ca^2+^-sensitive fluorescent dye (Rhod-5N) and with a confocal laser-scanning microscope to detect changes in the fluorescence emission of the cationic JC-1 dye, which changes its emission in response to variations in mitochondrial membrane potential. The changes observed suggested that capsaicin modifies mitochondrial Ca^2+^ and its membrane potential, presumably through TRPV1 channels localized to this compartment [49]. Interestingly, microglia derived from TRPV1-KO mice and treated with capsaicin displayed no changes in both parameters, indicating that the effects observed in the WT mice require the expression of TRPV1 channels in these cells. The cellular effects that arise from changes in Ca^2+^ levels in the mitochondria of these cells yield reactive oxygen species (mitROS) from this compartment which, in turn, act on the mitogen-activated protein kinase (MAPK) signaling pathway to induce microglial migration [49]. This exemplifies how the intracellular activity of TRPV1 influences cell processes such as cell migration.

Similarly, primary neonatal cardiomyocytes analyzed by confocal immunofluorescence and electron microscopy have revealed that these channels are located in the mitochondria, while they are scarce in the cell surface. At the same time, it has been reported that TRPV1 activation in these cells decreases the fluorescence intensity from mitochondria labeled with tetramethylrhodamine ethyl ester (TMRE; an indicator of active mitochondria) in cell imaging experiments, indicating changes in the mitochondrial membrane potential of these cells [51]. This effect was prevented by the co-treatment with capsaicin and capsazepine (a TRPV1 antagonist), showing that TRPV1 activation modifies mitochondrial function. Unexpectedly, this effect was also prevented by cyclosporine A, which is a calcineurin inhibitor. Calcineurin is a crucial mediator of TRPV1 desensitization [35], and Hurt and collaborators have reported that calcineurin directly interacts within the TRPV1 C-terminus, favoring channel dephosphorylation and providing negative feedback regulation through calmodulin [51]. Additionally, TRPV1 dephosphorylation through calcineurin’s actions primes the channel to be reactivated by recovering its phosphorylated state through actions of Ca^2+^ calmodulin-dependent kinase II (CaMKII) [63]. Thus, the cyclosporine A effects on cardiomyocytes where mitochondrial membrane potential decreases, i.e., as produced by capsaicin, could be caused through the disruption of the dephosphorylation/phosphorylation cycle of the TRPV1 channel.

Furthermore, using a myocardial infarction rodent model, Hurt and collaborators demonstrated that capsaicin decreases the infarct area, an effect that was also blocked by capsazepine and by cyclosporine A, suggesting that TRPV1 activation produces protective effects against reperfusion injury in this myocardial infarction model [51].

Additionally, Lang and collaborators have also identified the TRPV1 in cardiac mitochondrial protein fractions from WT mice [62] and have reported that mice fed with a chronic high salt diet display mitochondrial dysfunction, which is ameliorated by the concomitant administration of this diet and capsaicin, and not when capsaicin is co-administered to the TRPV1-KO mice, which confirmed that TRPV1 activation diminishes mitochondrial dysfunction produced by this type of diet. Remarkably, in mice treated with capsaicin, cardiac mitochondrial enzyme function and expression were restored (i.e., Sirtuin 3), suggesting that TRPV1 improves cardiac dysfunction [62].

Furthermore, using a cardiovascular disease mice model in *Apoe^−/−^* mice fed with a high fat diet, mitochondrial dysfunction and cardiac injury were also observed, but these alterations were partially prevented by co-feeding them with capsaicin. This protective effect is not observed in the double knockout *Apoe^−/−^/Trpv1^−/−^* mice, suggesting that TRPV1 channels are required for the improvement of heart function by capsaicin [64]. These results reinforced the notion that TRPV1 offers heart protective roles in different animal models displaying cardiac mitochondrial dysfunction. Similar effects have also been detected in the H9C2 cell line (rat heart tissue-derived cardiac myoblast cell line), because when these cells are subjected to ischemia–reoxygenation processes and treated with low capsaicin concentrations, they display an improvement in cell damage [51].

In contrast, other authors have reported that TRPV1 activation produces harmful effects for mitochondria and cardiac tissue. For instance, it was reported that H9C2 cells exposed to hypoxia–reoxygenation (H–R) conditions show reduced cell viability, which is potentiated by a high dose of capsaicin. This effect is achieved by increasing intracellular Ca^2+^, reducing the mitochondrial membrane potential and elevating mitochondrial superoxide production, leading to apoptosis of these cells [54]. Importantly, all of these effects are potentiated by capsaicin and attenuated by capsazepine. These evidences suggest that there are dose-dependent effects for capsaicin and that this compound promotes dual effects by either preventing or intensifying mitochondrial dysfunction in H9C2 cells depending on the concentration of vanilloid used in the experiments. Curiously, these cells display low TRPV1 expression, with channels mainly located at the ER where their activation increases cytosolic and mitochondrial Ca^2+^ levels, suggesting that TRPV1 channels localized at the ER could work as Ca^2+^ exchangers between the ER and the mitochondria [65], a role that needs to be studied in further detail. In addition, H9C2 cells exposed to H–R conditions display increases in TRPV1 phosphorylation, indicating channel sensitization. This finding is in agreement with a previous report performed using DRG neurons exposed to hypoxia conditions [66], where channel phosphorylation of serine residues increased due to the induction of hypoxia-inducible factor-1 alpha (HIF-1α). The latter regulates protein kinase C epsilon (PKCε) translocation to the plasma membrane, then this kinase phosphorylates serine 800 located at the TRPV1 C-terminus [66]. This evidence reinforces that the TRPV1 channel is sensitized by hypoxic conditions in different cell contexts, suggesting that capsaicin produces a synergic effect on a channel previously sensitized by an H–R condition in H9C2 cells, resulting in cell death.

In support of the evidence provided here for the effects of TRPV1 activation on mitochondrial dysfunction, some ischemia–reperfusion experiments performed in ex vivo hearts from WT and TRPV1-KO mice have revealed that the TRPV1-lacking hearts show improvement in cardiac function, suggesting that TRPV1 channels play an essential role in reperfusion injury [54]. In addition, the recent characterization of TRPV1 expression in endothelial cells and its intracellular localization has allowed us for the reinforcement of its role in endothelial dysfunction. Human endothelial cells treated with 12-(S)-HETE (an arachidonic acid metabolite and a partial agonist of TRPV1 [30]) display increased mitochondrial Ca^2+^, reduced oxygen consumption rates and decreased mitochondrial membrane potentials, all of which are indicators of mitochondrial dysfunction. These indicators are also present when human endothelial cells are treated with capsaicin [57]. Interestingly, streptozotocin-induced diabetic mice exhibit increased 12-(S)-HETE levels and show impaired vascular regeneration, while in TRPV1-KO mice treated with streptozotocin, there is protection against this impairment even when their 12-(S)-HETE levels are also increased. These results suggest that TRPV1 activation by 12-(S)-HETE in diabetic mice deteriorates mitochondrial and endothelial function, favoring vascular disease [57].

Similarly, it has been reported that 13-(S)-HODE (an endogenous TRPV1 agonist derived from linoleic acid [32]) causes mitochondrial dysfunction in airway epithelial cells through TRPV1 activation, which induces epithelial injury [67]. It was demonstrated in human bronchial epithelial cells exposed to 13-(S)-HODE that these cells exhibited mitochondrial Ca^2+^ overload and enhanced cytoplasmatic cytochrome C. Mitochondrial alterations were partially prevented by capsazepine or by using a siRNA to downregulate TRPV1 expression [67]. Interestingly, patients with atopic asthma display high levels of 13-(S)-HODE in sera, bronchoalveolar lavage, and sputum supernatants [67]. Thus, it is feasible to hypothesize that this fatty acid activates TRPV1 channels expressed in the airway epithelia, exacerbating mitochondrial dysfunction. Hence, TRPV1 and enzymes that produce this fatty acid (lipoxygenase) could be targets for the treatment of airway pathologies such as atopic asthma.

### 2.4. TRPV1 Channels and Mitochondrial Dysfunction in Neuronal Cells

Until now, there has been scarce evidence showing mitochondrial localization for TRPV1 in DRG neurons. In contrast, there are data supporting the TRPV1 roles in mitochondrial Ca^2+^ homeostasis in these cells, affecting cell viability, as has been determined in primary DRG neuronal cultures treated with resiniferatoxin and capsaicin [46,68,69]. These neurons display severe mitochondrial fragmentation after treatment with the aforementioned compounds, indicating that Ca^2+^ influx through TRPV1 channels produced organelle disruption [46]. Other reports have confirmed that DRG neurons treated with high concentrations of capsaicin or prolonged exposure to vanilloids display disturbances in their mitochondrial transmembrane potential and apoptotic cell death occurs [68] (Figure 3a).

We have to remember that the primordial functions of mitochondria are the production of energy, the scavenging of reactive oxygen species (ROS, which are co-produced with ATP), and buffering cytoplasmatic Ca^2+^ overload. The latter results are relevant to neuronal Ca^2+^ homeostasis, and require coordinated actions between the mitochondrial Ca^2+^ uniporter (MCU) that takes up cytoplasmic Ca^2+^, the Na^+^/Ca^2+^/Li^2+^ exchanger (NCLX) that mediates sodium-dependent Ca^2+^ efflux, and the permeability transition pore (PTP). The overload of mitochondrial Ca^2+^ uptake causes the opening of the PTP, releasing cytochrome C and inducing apoptosis [70] (Figure 3a). Interestingly, several reports have shown that mitochondria are also essential players in the pain response, since they can regulate the duration of the harmful signal transmission [71]. These effects have also been linked to the activation of TRPV1.

The toxic effects produced by mitochondrial Ca^2+^ overload have been studied in DRG neurons dissected from a rat spinal cord injury model. These neurons display decreased mitochondrial membrane potentials, and concomitantly increased ROS production and activation of caspases 3 and 9, compromising neuron viability and leading to apoptosis [72]. These effects are partially dependent upon TRPV1 activation [72].

Mitochondrial Ca^2+^ overload and cell death in DRG neurons treated with vanilloids seem to be specific for TRPV1, since activation of TRPA1 does not produce this overload even if cytoplasmatic Ca^2+^ increases [69], reinforcing the specificity of TRPV1′s effects on mobilization of mitochondrial Ca^2+^. These results were confirmed through experiments performed in TRPV1-expressing HEK293 cells, where capsaicin treatment modified mitochondrial potential and caused cell death. Notably, TRPV1 activation by extracellular acid pH in these cells does not affect mitochondrial function nor cell viability, suggesting that cell lethality by TRPV1 activation depends on the type of agonist present [69]. It could also indicate that protons cannot activate intracellularly located channels, so Ca^2+^ mobilization is not internally magnified, preventing cell death. Moreover, experiments performed with TRPV1-transfected HeLa cells have revealed that treatments with low capsaicin concentrations produced changes in the plasma membrane’s integrity, leading to necrotic cell death. In contrast, cells treated with high capsaicin concentrations show mitochondrial dysfunction and apoptotic cell death [73] (Figure 3a). Together, this evidence suggests that different doses of capsaicin trigger distinct molecular pathways to induce the same result: cell death.

The toxic effects of vanilloids have also been evidenced in neurons from the central nervous system [74]. For example, the identification of TRPV1 expression in dopaminergic neurons allowed for the exploration of the effects of TRPV1 agonists in these cells, demonstrating that high doses of capsaicin induce mitochondrial dysfunction and cell death. Similarly, anandamide, an endogenous agonist of the cannabinoid receptor 1 (CB1) and of TRPV1, causes these effects by activating both receptors, suggesting that an excessive local production of anandamide or any endogenous agonist of these receptors could induce neuronal damage through these proteins [74]. Cell death-inducing TRPV1 effects have also been observed in the neuronal TRPV1-transfected cell line, SH-SY5Y. The treatment of these cells with capsaicin or with the endogenous TRPV1 agonist, N-arachidonoyl-dopamine (NADA), or with anandamide (AEA), produced apoptotic cell death [75] (Figure 3a).

Additionally, it has been demonstrated that capsaicin produces axonal degeneration in sensory neurons through a mechanism that affects mitochondrial dynamics. Ca^2+^ overload produced by TRPV1′s activation induces axonal swelling and mitochondrial fission, blocking axonal regeneration [76] (Figure 3b). Remarkably, this could have two effects, depending on the type of neurons involved. For instance, in sensory neurons, axonal degeneration could produce neuronal dysfunction resulting in the block of the transduction and transmission of noxious stimuli, yielding an analgesic effect. In contrast, effects of capsaicin on neurons from the central nervous system could produce cell death, leading to neurodegenerative diseases [76].

The evidence described above suggests that TRPV1 activation causes mitochondrial Ca^2+^ unbalance, disrupting the mitochondrial membrane potential, inducing increased ROS production and cytochrome C release and leading to apoptosis (Figure 3a). TRPV1 activation and Ca^2+^ overload are not always harmful to mitochondria; some reports have also shown that a rise in Ca^2+^ can be buffered by this organelle and serve as a regulator of the painful response. For instance, DRG neurons exposed to capsaicin exhibit fast Ca^2+^ intracellular rises, which are buffered by their mitochondria. Next, Ca^2+^ is slowly released from this organelle, ensuring the maintenance of elevated cytosolic Ca^2+^ and, in parallel, TRPV1 desensitization could also occur [77].

Similarly, it has been demonstrated that knockdown of NCLX in DRG neurons decreases capsaicin-evoked action potentials. These neurons maintain the increase of cytosolic Ca^2+^ produced through TRPV1 activation, facilitating channel desensitization [78] (Figure 3c). These data reinforce the idea that intracellular Ca^2+^ overloads produced by TRPV1 activation are buffered by mitochondrial transporters, resulting in the modulation of TRPV1 function and leading to the regulation of the way noxious signals are transduced through this channel [78].

Furthermore, experiments performed with DRG and spinal cord neuron co-cultures evidenced that a sustained mitochondrial Ca^2+^ mobilization facilitates neurotransmitter release from presynaptic neurons, inducing the firing of action potentials from postsynaptic neurons. Remarkably, the duration of postsynaptic activity depends on presynaptic stimuli strength, where an intense stimulation (such as a high capsaicin concentration) induces long-lasting postsynaptic activity. This effect is independent of extracellular Ca^2+^, but it depends on the proper function of the presynaptic mitochondria [79].

Sensory neurons display numerous mitochondria located in the synaptic buttons, providing energy and clearing intracellular Ca^2+^ from presynaptic neurons [80]. Thus, mitochondria provide a long-lasting effect of pain signal transmission by dodging the Ca^2+^-dependent desensitization of some ion channels, such as TRPV1, and this exemplifies the close relationship between mitochondrial Ca^2+^ mobilization and the maintenance of noxious signal transmission so that a sustained pain response can be produced.

## 3. Conclusions

Mitochondria are organelles that not only function as producers of cellular ATP, but also perform essential roles in prompting apoptotic cell death. Many apoptotic stimuli can lead to the liberation of specific mitochondrial pro-apoptotic factors into the cytosol and, although the molecular mechanism underlying this is still under debate, mitochondrial Ca^2+^ overload is definitely one of the signals involved in swelling of mitochondria, rupture of their outer membranes, and release of mitochondrial apoptotic factors into the cytosol.

As a result of this mitochondrial dysfunction, disorders such as those of an inflammatory nociceptive nature where there is sensory nerve hyperexcitability are also produced. Signals that produce mitochondrial dysfunction result in the production of ROS, mitochondrial depolarization and Ca^2+^ release, regulating cellular signaling. A link between this and nociception has been highlighted in the last few years and TRPV1 has emerged as an important effector of this through the regulation of mitochondrial function.

Several reports have evidenced how TRPV1 activation modifies mitochondrial function in non-neuronal and neuronal cells. There are still several questions that must be answered in relation to mitochondrial TRPV1 localization, although most of the studies described here show that TRPV1 activation has a pivotal influence on mitochondrial Ca^2+^ homeostasis, the mitochondrial channel localization is still a fertile field to explore.

This review summarizes the works describing TRPV1 protein detection in the mitochondrial fraction of cardiac tissue and also the reports showing channel expression in primary neonatal cardiomyocytes, microglia, and the H9C2 cell line [49,51,54]. The localization of this channel in the mitochondria has been based on the co-localization of the channel with specific mitochondrial markers. However, these studies have not discerned if the TRPV1 signal observed arises from the channel located in the mitochondria-associated ER membranes (MAM), which are microdomains of the ER with close proximity to the mitochondria. It is possible to speculate the latter, since it has recently been characterized that TRPV1 directly interacts with the Sigma-1 receptor [81], a protein highly enriched in the MAM [82], and implicated in the regulation of interorganellar Ca^2+^ between ER and mitochondria [82].

We also speculate that TRPV1 could associate to the mitochondrial membrane; nonetheless, the only approach that has suggested this is one where electron microscopy was used, and where it was shown that immunogold labeling of TRPV1 displayed an irregular distribution in the mitochondria from neonatal cardiomyocytes. Hence, mitochondrial channel distribution in these cells was not limited to the mitochondrial membrane as could be hypothesized.

These works reveal that it is still necessary to conduct in-depth tests to clarify if TRPV1 channels are really located in the mitochondrial membrane and to elucidate the topological orientation of the channel in this organelle. Moreover, it will be interesting to study isolated mitochondria from non-neuronal and neuronal cells to determine if TRPV1 is directly located in the mitochondria and moreover if capsaicin treatment of these isolated mitochondria could directly modify the membrane potential of this organelle (i.e., electrophysiological recordings).

From the advances we have observed in the last few years, the field of study of the role of proteins such as TRPV1 in mitochondrial function is one that promises to yield rich information on how apoptosis and/or pain are regulated by ion channels at the subcellular level.

## Figures and Tables

**Figure 1 ijms-21-08882-f001:**
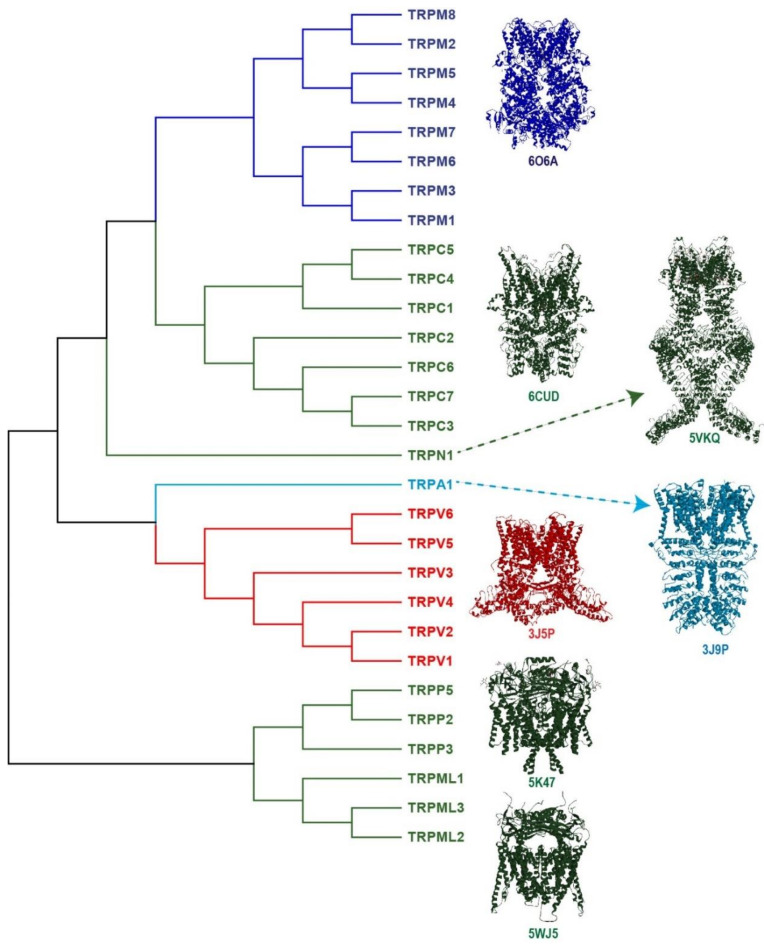
The Transient Receptor Potential (TRP) superfamily of ion channels. TRP ion channels are classified into seven subfamilies: TRPC (canonical), TRPV (vanilloid), TRPM (melastatin), TRPP (polycystin), TRPML (mucolipin), TRPA1 (ankyrin) and TRPN (no mechanoreceptor). The figure shows the phylogenetic analysis between the human TRP protein sequences; the alignment was obtained using ClustalW2 at the EMBL-EBI server. For TRPC2 (pseudogene) and TRPN (not expressed in humans), the sequences used were from mice and fish, respectively. The channels colored in dark-, light-blue and red refer to some of the channels known as the thermo-TRP, since there are activated by cool, cold and hot temperatures (TRPM81, TRPA1 and TRPV1, respectively). The figure also shows the tetrameric 3D-structure of a representative member of each subfamily. TRPM8 (6O6A), TRPC3 (6CUD), TRPN (5VKQ), TRPA1 (3J9P), TRPV1 (3J5P) TRPP2 (5K47) and TRPML1 (5WJ5). The symbols beneath each 3D-structure are the number access in the Protein Data Bank (PDB).

**Figure 2 ijms-21-08882-f002:**
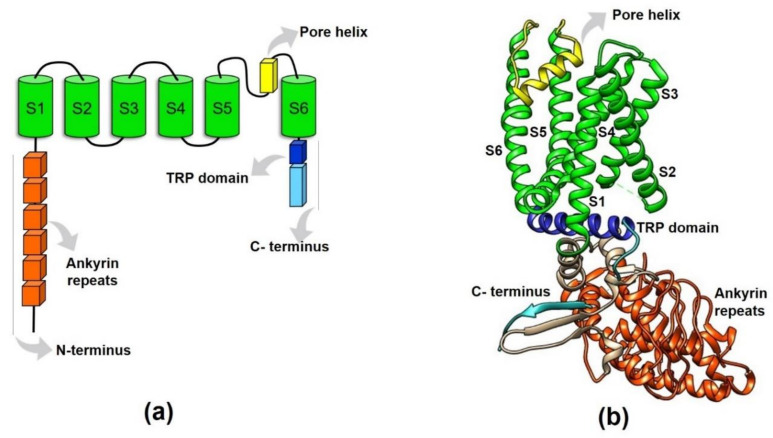
Structural features of the TRPV1 protein. (**a**) This figure schematizes the domains of a TRPV1 monomer (or subunit). The amino- and carboxy- termini are intracellularly located (N- and C- termini), the six ankyrin repeats and the TRP-box are contained in the N- and C-termini, respectively. The re-entrant loop between the S5–S6 forms the ionic conduction pore when the tetramer is formed. (**b**) 3D-representation of a TRPV1 subunit displaying the arrangement of the domains represented in (**a**). S1–S6: transmembrane segments.

**Figure 3 ijms-21-08882-f003:**
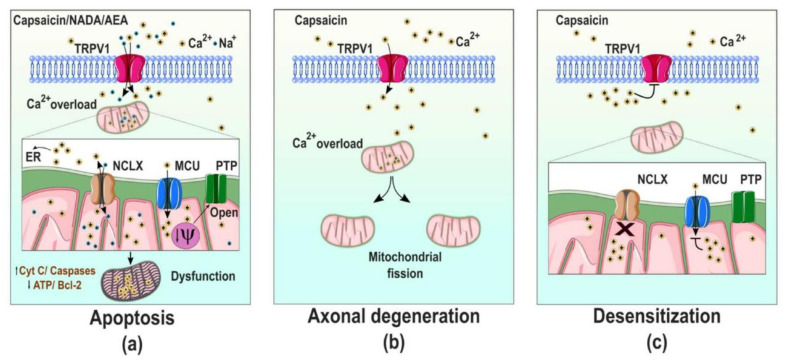
Scheme representing plasmatic TRPV1 actions on mitochondria. (**a**) Prolonged activation of TRPV1 leads to Ca^2+^ overload into the mitochondria. This effect triggers mitochondrial depolarization, releases cytochrome C through the permeability transition pore (PTP) and induces apoptosis. (**b**) Activation of TRPV1 induces the mitochondrial fission and produces axonal degeneration. (**c**) Downregulation of the Na^+^/Ca^2+^/Li^2+^ exchanger (NCLX) expression avoids the release of Ca^2+^ from the mitochondria, which inhibits mitochondrial Ca^2+^ uptake through the mitochondrial Ca^2+^ uniporter (MCU), leading to accumulation of cytosolic Ca^2+^ and inducing TRPV1 desensitization. ER: endoplasmic reticulum. AEA: N-arachidonoylethanolamine or Anandamide. NADA: N-Arachidonoyl-Dopamine.

**Table 1 ijms-21-08882-t001:** Intracellular localization of the TRPV1 channels in different cell types. DRG: dorsal root ganglion.

Intracellular Localization	Cell Type	Reference
Mitochondria	Murine cardiomyocytes	[51]
	Microglial cells	[49]
	Heart-derived H9C2 cells	[51,54]
	Human parietal cells	[55]
	Human odontoblast-like cells	[56]
	Human endothelial cells	[57]
Endoplasmic Reticulum	DRG neurons	[44,45,46,47,48]
	Airway smooth muscle cells	[50]
	TRPV1-expressing cells: HEK293, COS7, Sf9, HeLa	[44,45,46,47,52,58]
Sarcoplasmic Reticulum	Skeletal muscle cells	[59,60]
Golgi Complex	DRG neurons	[24]
	Microglia cells	[49]
	Breast cancer cell lines	[61]
Lysosome	Microglia cells	[49]

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
