# Peer review of "TRPV1 Channel: A Noxious Signal Transducer That Affects Mitochondrial Function"

_ijms, 2020, doi:10.3390/ijms21238882_

Round 1

Reviewer 1 Report

In this review, Juárez-Contreras et al. elegantly described the role of TRPV1 in the mitochondria. In general, this is a well-written review that enables the reader to learn the state of the art of our knowledge regarding the TRPV1 role in intracellular organelles. Thus, this review is of significant importance. The only critique I have is that the authors did not mention the scientific/experimental level of the different studies they referenced. Hence, the impression may be that all the information provided in this review should be regarded as solid and well-proven. Unfortunately, this is not the case, and in several studies repeatedly cited in this review, there are several gaps of knowledge, which should be noted in my point of view. However, if the authors feel that such a critique is not suitable, they should mention how many different labs had described a specific phenomenon. This will enable the reader to judge by itself the robustness of the results.     

Author Response

In this review, Juárez-Contreras et al. elegantly described the role of TRPV1 in the mitochondria. In general, this is a well-written review that enables the reader to learn the state of the art of our knowledge regarding the TRPV1 role in intracellular organelles. Thus, this review is of significant importance. The only critique I have is that the authors did not mention the scientific/experimental level of the different studies they referenced. Hence, the impression may be that all the information provided in this review should be regarded as solid and well-proven. Unfortunately, this is not the case, and in several studies repeatedly cited in this review, there are several gaps of knowledge, which should be noted in my point of view. However, if the authors feel that such a critique is not suitable, they should mention how many different labs had described a specific phenomenon. This will enable the reader to judge by itself the robustness of the results. 

We thank this reviewer for her/his valuable comments that have strengthened our manuscript.

We have attended your recommendations and now we discuss the robustness of the results described in our review at the end of the conclusion section. The added paragraphs read as follows:

“Several reports have evidenced how TRPV1 activation modifies mitochondrial function in non-neuronal and neuronal cells. There are still several questions to answer related to mitochondrial TRPV1 localization, although, most of the studies described here show that TRPV1 activation has a pivotal influence on mitochondrial Ca2+ homeostasis, while mitochondrial channel localization is still a fertile field to explore.

This review summarized the works describing TRPV1 protein detection in the mitochondrial fraction of cardiac tissue and also the reports showing channel expression in primary neonatal cardiomyocytes, microglia and the H9C2 cell line [49, 51, 54]. The localization of this channel in the mitochondria has been based in the colocalization of the channel with specific mitochondrial markers. However, these studies have not discerned if the TRPV1 signal observed arises from the channel located in the mitochondria-associated ER membrane (MAM), which are microdomains of the ER with close proximity with the mitochondria. This speculation could be possible since it has recently been characterized that TRPV1 directly interacts with the Sigma-1 receptor [81], a protein highly enriched in the MAM [82], and implied in the regulation of interorganellar Ca2+ between ER and mitochondria [82].

We may also speculate that TRPV1 could associate to the mitochondrial membrane; nonetheless, the only approach that has suggested this it one where electron microscopy was used and where it was shown that immunogold labeling of TRPV1 displayed an irregular distribution in the mitochondria from neonatal cardiomyocytes. Hence, mitochondrial channel distribution in these cells was not limited to the mitochondrial membrane as we could hypothesize.

These evidences reveal that it is still necessary to conduct in-depth tests to clarify if TRPV1 channels are really located in the mitochondrial membrane and to elucidate the topological orientation of the channel in this organelle. Moreover, it will be interesting to study isolated mitochondria from non-neuronal and neuronal cells to determine if TRPV1 is directly located in the mitochondria and moreover if capsaicin treatment of these isolated mitochondria could directly modify the membrane potential of this organelle (i.e. electrophysiological recordings). 

From the advances we have observed in the last few years, the field of study of the role of proteins such as TRPV1 in mitochondrial function is one that promises to yield rich information on how apoptosis and/or pain are regulated by ion channels at the subcellular level”.

Additionally, we have modified the title and reorganized the sections in order to highlight that TRPV1 activation can regulate mitochondrial Ca2+ homeostasis independently of whether the channel is located in the cell surface or in intracellular membranes.

Reviewer 2 Report

This review by Rebeca et al. try to summarize the link of TRPV1 activation and mitochondrial function. They seems want to emphasize the importance of mitochondria localized TRPV1, but the description are confusing most of time. 

Major problem: 

  1. Logically, in 2.2 the authors highlight the intracellular TRPV1 localization, however, there is almost nothing to do with the intracellular mitochondrial TRPV1in 2.3, or the evidence is weak. It seems the effects descripted in the Fig3 are mainly through plasma membrane TRPV1 but not mitochondrial TRPV1. Maybe those references have the evidence, but the reviewers didn't do a good job to summarize. I didn't see how they adress what they said in line 22-23 "highlighting pivotal intracellular actions for this ion channel" or in line 83-85 (Sentence 83-85 is also too long to read...)
  2.   If the authors want to highligt the mitochondrial TRPV1, they need to explain more details of TRPV1 in mitochontria, such as the specific localizaion, the orientation, the gating mechanism. But TRPV1 is not even pictured in the mitochondria in Fig3...
  3. In line with the above problems, the authors write too much on the mitochondria, but not really emphasize TRPV1. For example, in the final conclusion 351-365, it's all well known importance of mitochondria. 

Minor problem:

  1. The title is misleading, the authos must also agree TRPV1 is not only a channel in the mitochondria.
  2. Fig 1 is pretty but not so meaningful. A TRP channel fammily tree would be more suitable. 
  3. Citation needed on line 121.
  4. Table 1 can be more specific, for example listing which paper is for microglia. And in the paragraph under table 1, the authors mentioned the Sarcoplasmic reticulum, the reference [52], but it's not in Table1.
  5. Again for Table 1, the authors mentioned lysosome in ling 289, but not listed in Table 1?
  6. Figure 3c is misleading, does it apply for the desensitization written in line 260??
  7. In line 226, it says TRPV1 agonist triggers a same result cell death. However, in 218-219 it says TRPV1 activation by protons will not do so.  

Author Response

This review by Rebeca et al. try to summarize the link of TRPV1 activation and mitochondrial function. They seems want to emphasize the importance of mitochondria localized TRPV1, but the description are confusing most of time. 

Major problem: 

  1. Logically, in 2.2 the authors highlight the intracellular TRPV1 localization, however, there is almost nothing to do with the intracellular mitochondrial TRPV1in 2.3, or the evidence is weak.

We thank this reviewer for valuable insights that have helped improve our manuscript.

Indeed, the evidence suggesting mitochondrial localization in neurons is null: channel localization in this organelle only has been demonstrated in non-neuronal cells (as we mentioned in section 2.4).

To clarify this point, we have made two changes:

  1. We have modified the title of our review; now it reads:

 “TRPV1 channel: a noxious signal transducer that affects mitochondrial function” 

We consider that this title allows for the two following possibilities: to understand that TRPV1 effects are through channels located in the mitochondria (as what happens in non-neuronal cells) or that the effects occur through channels located at the cell surface (in neurons).

  1. We have changed the introductory paragraph to section 2.3 (now section 2.4) to emphasize the lack of evidence for mitochondrial localization of TRPV1 in neurons. Now the first paragraph reads:

“Until now, there is scarce evidence showing mitochondrial localization for TRPV1 in DRG neurons. In contrast, several data support roles for TRPV1 in mitochondrial Ca2+ homeostasis in these cells, affecting cell viability, as has been determined in primary DRG neuronal cultures treated with resiniferatoxin and capsaicin [46, 68, 69]. These neurons display severe mitochondrial fragmentation after the treatment with the aforementioned compounds, indicating that Ca2+ influx through TRPV1 channels produced organelle disruption [46]. Other reports have confirmed that DRG neurons treated with high concentrations of capsaicin or prolonged exposure to vanilloids display disturbance in their mitochondrial transmembrane potential and apoptotic cell death occurs [68], Figure 3a.”

  • It seems the effects descripted in the Fig3 are mainly through plasma membrane TRPV1 but not mitochondrial TRPV1. Maybe those references have the evidence, but the reviewers didn't do a good job to summarize.

Yes, you are correct. Figure 3 shows that mitochondrial dysfunction is produced by the activation of channels located in the cell surface. The original references have not shown that these channels are located in the mitochondria; thus, we do not mention that; we only describe the effects of TRPV1 agonists on mitochondrial function.

I didn't see how they adress what they said in line 22-23 "highlighting pivotal intracellular actions for this ion channel" or in line 83-85 (Sentence 83-85 is also too long to read...)

Thanks for the observation: The experimental evidence shows that TRPV1 activation affects calcium homeostasis in cells where the channels have been located in this organelle (some non-neuronal cells) and neurons that do not show this intracellular localization. To avoid reader confusion, we have re-written the lines 22-23 (now 21-22) that you mention:

21-22 lines now read:

“This review focuses on describing experimental evidence showing that TRPV1 influences mitochondrial function.”

98-101 lines (83-85) lines, now read:

“In this review, we will describe how TRPV1 activation modifies mitochondrial function. Here, we will discuss the data showing that the channel's activation can induce mitochondrial dysfunction in neuronal and non-neuronal cells”.

If the authors want to highligt the mitochondrial TRPV1, they need to explain more details of TRPV1 in mitochontria, such as the specific localizaion, the orientation, the gating mechanism. But TRPV1 is not even pictured in the mitochondria in Fig3...

We have not described these details because there is no experimental evidence about them; thus, it is an interesting open question in the TRPV1 channel field of research .However, we did add the words “plasmatic TRPV1” to the figure legend title (line 379).

In line with the above problems, the authors write too much on the mitochondria, but not really emphasize TRPV1. For example, in the final conclusion 351-365, it's all well known importance of mitochondria. 

We have performed your suggestion and now the conclusion section reads:

“Mitochondria are organelles that not only function as producers of cellular ATP but they also perform essential roles in prompting apoptotic cell death. Many apoptotic stimuli can lead to the liberation of specific mitochondrial pro-apoptotic factors into the cytosol and, although the molecular mechanism underlying this is still under debate, mitochondrial Ca2+ overload is definitely one of the signals involved in swelling of mitochondria, rupture of their outer membranes and release of mitochondrial apoptotic factors into the cytosol.

As a result of this mitochondrial dysfunction, disorders such as those of an inflammatory nociceptive nature where there is sensory nerve hyperexcitability are also produced. Signals that produce mitochondrial dysfunction result in the production of ROS, mitochondrial depolarization and Ca2+ release, regulating cellular signaling. A link between this and nociception has been highlighted in the last few years and TRPV1 has emerged as an important effector of this through the regulation of mitochondrial function.

Several reports have evidenced how TRPV1 activation modifies mitochondrial function in non-neuronal and neuronal cells. There are still several questions to answer related to mitochondrial TRPV1 localization, although, most of the studies described here show that TRPV1 activation has a pivotal influence on mitochondrial Ca2+ homeostasis, while mitochondrial channel localization is still a fertile field to explore.

This review summarized the works describing TRPV1 protein detection in the mitochondrial fraction of cardiac tissue and also the reports showing channel expression in primary neonatal cardiomyocytes, microglia and the H9C2 cell line [49, 51, 54]. The localization of this channel in the mitochondria has been based in the colocalization of the channel with specific mitochondrial markers. However, these studies have not discerned if the TRPV1 signal observed arises from the channel located in the mitochondria-associated ER membrane (MAM), which are microdomains of the ER with close proximity with the mitochondria. This speculation could be possible since it has recently been characterized that TRPV1 directly interacts with the Sigma-1 receptor [81], a protein highly enriched in the MAM [82], and implied in the regulation of interorganellar Ca2+ between ER and mitochondria [82].

We may also speculate that TRPV1 could associate to the mitochondrial membrane; nonetheless, the only approach that has suggested this it one where electron microscopy was used and where it was shown that immunogold labeling of TRPV1 displayed an irregular distribution in the mitochondria from neonatal cardiomyocytes. Hence, mitochondrial channel distribution in these cells was not limited to the mitochondrial membrane as we could hypothesize.

These evidences reveal that it is still necessary to conduct in-depth tests to clarify if TRPV1 channels are really located in the mitochondrial membrane and to elucidate the topological orientation of the channel in this organelle. Moreover, it will be interesting to study isolated mitochondria from non-neuronal and neuronal cells to determine if TRPV1 is directly located in the mitochondria and moreover if capsaicin treatment of these isolated mitochondria could directly modify the membrane potential of this organelle (i.e. electrophysiological recordings). 

From the advances we have observed in the last few years, the field of study of the role of proteins such as TRPV1 in mitochondrial function is one that promises to yield rich information on how apoptosis and/or pain are regulated by ion channels at the subcellular level.”

Minor problem:

The title is misleading, the authos must also agree TRPV1 is not only a channel in the mitochondria.

Thanks for the suggestion; we agree with you, now the title has been modified.

Fig 1 is pretty but not so meaningful. A TRP channel fammily tree would be more suitable. 

We have changed the figure as you suggested.

Citation needed on line 121.

We have added the missing reference. (line 154).

Table 1 can be more specific, for example listing which paper is for microglia. And in the paragraph under table 1, the authors mentioned the Sarcoplasmic reticulum, the reference [52], but it's not in Table1.

Thanks for the comment, the table has been modified as you suggested.

Again for Table 1, the authors mentioned lysosome in ling 289, but not listed in Table 1?

This has also been fixed.

Figure 3c is misleading, does it apply for the desensitization written in line 260??

The comment is correct, actually, the figure refers to the next paragraph:

“Similarly, it has been demonstrated that knockdown of NCLX in DRG neurons decreases capsaicin evoked action potentials. These neurons maintain the increase of cytosolic Ca2+ produced through TRPV1 activation, facilitating channel desensitization [78], Figure 3c.” (lines 412-414).

In line 226, it says TRPV1 agonist triggers a same result cell death. However, in 218-219 it says TRPV1 activation by protons will not do so.  

Thanks for the observation the lines refer to capsaicin, we change this to avoid confusion. Now the text reads:

Together, these evidences suggest that different doses of capsaicin trigger distinct molecular pathways to induce the same result: cell death.”. (lines 375-376)

Additionally, we have modified the title and reorganized the sections in order to highlight that TRPV1 activation can regulate mitochondrial Ca2+ homeostasis independently of whether the channel is located in the cell surface or in intracellular membranes.

Reviewer 3 Report

Abstract

Line 18:function"s"

Introduction

Line 32: a delayed response under prolonged bright…………

I am not sure whether the authors have indeed read through the reference (1), because the fig 3 of the reference 1 certainly demonstrated that the mutant showed a more slow or delay response when compared with the wild type. However, the fig1 of the reference 1 showed completely different voltages between the wild type and the mutant. The larger positive phase of the on-transiet was absent in the mutant. Results of the fig1 were not merely caused by the “delay response”. I may interpretate these data as the mutant presented changes of the voltages and response under light stimulation.

Line46: since you have clearly numbered the Calmodulin-binding sites, please do show the numbers of ankyrin-like repeats in the sentence.

Page 3 Fig 1

Please explain the meanings of the symbols beneath each 3 d structure of the TRP proteins, such as 6O6A?, 3J9P…………….

Line 113:sensibilization

I have gone through the reference 27. https://www.pnas.org/content/100/21/12480.long

There are only 4 times of sensitize in this paper, including one in the title and the rest 3 in the context. Unfortunately, the paper did not mention or use the term” sensibilization”. It will be practical to quote any paper to explain the term “sensibilization”. I also check the key words sensibilization, TRPV1 in the pubmed website, and the results were “zero results.” Please do check the term you used in this paper is corrective.

Line 127-140 a quite long sentence to compare TRPV1 with Kv channel. It will be good to have a “brief introduction regarding the “importance of Kv”.

Line 154-158 We can understand the biting sites for the capsaicin 511, 512, and 550 were in the inner side of the TRPV1 due to the nature of the capsacin. Arguably, I feel the viewpoints addressing here are too emphasizing the “structure”, instead of the”functions”. The way of introducing advanced findings regarding the ultrastructure of the TRPV1 here is too shallow.

Line 249 Des-functionalization? Reference 61 did not use “des-functionalization”. It will be practical to quote any paper to explain the term “Des-functionalization”. I also check the key words Des-functionalization, TRPV1 in the pubmed website, and the results were “zero results.” Please do check the term you used in this paper is corrective.

Line297-303. In the previous sentences you mentioned that TRPV1 activation leading to the damage of mitochondria. However, you stated that the TRPV1 has protective effects, and the effects are through phosphorylation, but you said that the protect effects are blocked by capsazepine and Cyclosporine. I suggest the authors should be carefully rewritten these sentences, because the content of these sentences may confuse your readers.

Line 317-324. H/R reduced cell viability, which is which is potentiated by a high dose of capsaicin. This effect is achieved by increasing intracellular Ca2+ and triggering apoptosis in these cells [67]. Unexpectedly, H9C2 cells under H/R conditions also exhibited TRPV1 phosphorylation and capsaicin resulted in increased mitochondrial dysfunction, which was inhibited by capsazepine.

These sentences only described parts of fact. Hypoxia is known to induced hypoxia-inducible factor.

Hypoxia can directly damage mitochondria through the lack of oxygen to impede electron transfer through the electron transport chain.

Hypoxia upregulates HIF, which then upregulate TRPV1. I presume that the functions of mitochondria successfully influenced by the TRPV1 antagoinist, suggesting that TRPV1 may be the “effector” under hypoxia. Several papers support this idea. Please pain 2011 apr 152 (4):936-945 International scholar research notices 2011 Yu, C.S. : study on HIF-1a gene translation in psoriatic epidermis with the tpical treatment of capsaicin ointment.

Have you ever thought of a question: whether the phosphorylation of TRPV1, may not come from kinases, but from mitochondria? Therefore, the interaction between TRPV1 and mitochondria, like symbiotic , work together in against disease or causing disease? 

Author Response

We thank this reviewer for their valuable comments that have strengthened our manuscript. Please find our answers to your concerns below.

Abstract

Line 18:function"s" .. We have fixed this.

Introduction

Line 32: a delayed response under prolonged bright…………

I am not sure whether the authors have indeed read through the reference (1), because the fig 3 of the reference 1 certainly demonstrated that the mutant showed a more slow or delay response when compared with the wild type. However, the fig1 of the reference 1 showed completely different voltages between the wild type and the mutant. The larger positive phase of the on-transiet was absent in the mutant. Results of the fig1 were not merely caused by the “delay response”. I may interpretate these data as the mutant presented changes of the voltages and response under light stimulation.

We have rewritten this sentence and added the following paragraphs (lines 30-77):

The first evidence for the existence of these channels was found in the Drosophila melanogaster fly. Cosens and Manning [1] analyzed a spontaneous mutant that displayed transient rather than sustained responses in the presence of prolonged bright illumination in electroretinogram (ERG) measurements [1]. This mutation rendered the flies blind and the investigators who isolated this mutant named it the "A-type" mutant, ascribing its phenotype to a failure of photopigment regeneration [2]. Nonetheless, several concerns were raised at the time of its isolation, with the main one being that the results were based on a single spontaneously occurring mutant, for which there was no description of its genetic background. Hence, there was a lack of knowledge on what genetic alterations this strain embodied with the results possibly being due to additive effects of alterations in several genes mapping to the same chromosome [3]. Pak and collaborators proved that the observed phenotype was indeed due to mutation in a single gene by isolating multiple mutated alleles from a baseline stock of known genetic background [3]. It was also important to establish the cellular origins of the measured ERG components because it had not been determined whether the lack of a sustained response seen in the ERG of this strain originated from the photoreceptors or from other retinal cells [3]. In 1975, Minke and collaborators performed intracellular recordings from the mutant photoreceptors and were able to determine that the defect arose from the photoreceptors. Thus, they concluded that this mutant is defective in phototransduction [4]. Several other studies were performed with this mutant helped to forge a representative name for this mutant: the “transient receptor potential” or trp [4-6]

Ten years later, the trp gene was isolated [7] and characterized as a genomic region that encodes for some RNA species missing in the mutant fly [8]. Interestingly, transformation of the mutant germline with the region encoding for the trp gene reestablished the normal phototransduction phenotype in the trp mutant fly, confirming the identity of this gene [7]. Finally, the cloning of the trp gene suggested a transmembrane protein with 8 α-helices containing highly hydrophilic N- and C-termini, which was consistent with already known receptor/transporter/channel proteins [9, 10].

The TRP protein was localized to the rhabdomere and hence thought to participate in phototransduction but the absence of homologous proteins to the TRP in available databases and other results that suggested that in null trp alleles there was still a persistent sustained receptor potential under dim light stimulation [4, 11], it was concluded that the trp gene does not encode for the light-sensitive channel [9, 10].

Later, it was shown that La3+ could mimic the trp phenotype [12] and this led to the proposal that the TRP encoded for an inositide-activated Ca2+ channel/transporter required for Ca2+ stores refilling [13].Voltage-clamp recording in the whole-cell configuration showed that the primary defect in the trp mutant was a drastic reduction in the Ca2+ permeability of the light-sensitive channels themselves [14].

Then, another protein similar to the trp gene product was found, the TRP-like (TRPL) [15]. The product encoded by the trpl gene exhibited characteristics of a transmembrane protein that contains two putative calmodulin-binding sites and an ankyrin-like repeat domain [15]. Sequence homology analyses demonstrated that the central transmembrane region of the trp and trpl products resemble the transmembrane segments (S1-S6) of calcium channels, suggesting that these TRP proteins could act as ion channels [15]. The functional description of invertebrate TRP proteins demonstrated that these proteins function as light-sensitive ion channels that allow calcium entry in Drosophila photoreceptors [14] and the phenotype of the trp mutation was reinterpreted and it was suggested that the light response of Drosophila is mediated by channels composed from the trp and trpl  gene products [14, 15]. In summary, there are two distinct conductances that give rise to the normal light-sensitive current: one encoded by the trp gene that is highly Ca2+ selective and a second channel that is responsible for the residual light-sensitive current in the trp mutants encoded by the homologous gene trpl.

since you have clearly numbered the Calmodulin-binding sites, please do show the numbers of ankyrin-like repeats in the sentence

The numbers of ankyrins-like domains are mentioned, and the lines read: “that contains two putative calmodulin-binding sites and an ankyrin-like repeat domain [15].”

Page 3 Fig 1

Please explain the meanings of the symbols beneath each 3D structure of the TRP proteins, such as 6O6A?, 3J9P…………….

The meanings of the symbols are now stated in the figure legend

Line 113:sensibilization

I have gone through the reference 27. https://www.pnas.org/content/100/21/12480.long

There are only 4 times of sensitize in this paper, including one in the title and the rest 3 in the context. Unfortunately, the paper did not mention or use the term” sensibilization”. It will be practical to quote any paper to explain the term “sensibilization”. I also check the key words sensibilization, TRPV1 in the pubmed website, and the results were “zero results.” Please do check the term you used in this paper is corrective.

We appreciate your comment, using the term sensibilization was a “translation” mistake, the correct word is sensitization as is mentioned in the original article paper that you have referred to (line 145).

Line 127-140 a quite long sentence to compare TRPV1 with Kv channel. It will be good to have a “brief introduction regarding the “importance of Kv”.

Thanks for your suggestion. In general, TRP channels resemble voltage-activated ion channels. In fact, our group suggested that TRPV1 could resemble a Kv channel in 2009 (doi: 10.4161/chan.3.5.9659). In addition, when the cryo-EM structure was resolved, it was also determined that this channel retains reminiscent structural similarity with Kv and Nav channels. We have rewritten the following sentence (lines 161-169) to further clarify this:

“This structure confirmed the tetrameric channel conformation with a large open basket-like domain corresponding to the intracellular N- and C-termini and a compact transmembrane domain that resembles that of transmembrane domain structure of Kv and Na channels [41]. TRP channels in general resemble voltage-activated ion channels. They display reminiscent structural features such as the presence of an antiparallel -sheet from the linker region and a -strand from the C-terminal region that make contact with two of the ankyrin repeats of an adjacent subunit (which presumably is important for channel assembly) [42]. These features are similar to the T1 domain of Kv channels where this region plays an important role in subunit assembly [43]. On the other hand, the wider outer pore and the shorter selectivity filter of TRPV1 also point to similarities with bacterial Nav channels [42]”

Line 154-158 We can understand the biting sites for the capsaicin 511, 512, and 550 were in the inner side of the TRPV1 due to the nature of the capsacin. Arguably, I feel the viewpoints addressing here are too emphasizing the “structure”, instead of the”functions”. The way of introducing advanced findings regarding the ultrastructure of the TRPV1 here is too shallow.

Thank for the comment, we agree with you since the main objective of this review is not to discuss ultrastructure of the TRPV1 channel in depth, we wanted to give general aspects, thus this part was eliminated.

Line 249 Des-functionalization? Reference 61 did not use “des-functionalization”. It will be practical to quote any paper to explain the term “Des-functionalization”. I also check the key words Des-functionalization, TRPV1 in the pubmed website, and the results were “zero results.” Please do check the term you used in this paper is corrective.

We appreciate your comment, the des-functionalization word was replaced by the word dysfunction (line 389), as presented in the original publication.

Line297-303. In the previous sentences you mentioned that TRPV1 activation leading to the damage of mitochondria. However, you stated that the TRPV1 has protective effects, and the effects are through phosphorylation, but you said that the protect effects are blocked by capsazepine and Cyclosporine. I suggest the authors should be carefully rewritten these sentences, because the content of these sentences may confuse your readers.

The paragraph was rewritten and now it reads (lines 253-272):

Similarly, primary neonatal cardiomyocytes analyzed by confocal immunofluorescence and electron microscopy have revealed that these channels are located in the mitochondria while they are scarce in the cell surface. At the same time, it has been reported that TRPV1 activation in these cells decreases the fluorescence intensity from mitochondria labeled with TMRE (tetramethylrhodamine, ethyl ester, an indicator of active mitochondria) in cell imaging experiments, indicating changes in the mitochondrial membrane potential of these cells [51]. This effect was prevented by the co-treatment with capsaicin and capsazepine (a TRPV1 antagonist), showing that TRPV1 activation modifies mitochondrial function. Unexpectedly, this effect was also prevented by cyclosporine A, which is a calcineurin inhibitor. Calcineurin is a crucial mediator of TRPV1 desensitization [35] and Hurt and collaborators have reported that calcineurin directly interacts within the TRPV1 C-terminus, favoring channel dephosphorylation and providing negative feedback regulation through calmodulin [51]. Additionally, TRPV1 dephosphorylation through calcineurin’s actions primes the channel to be reactivated by recovering its phosphorylated state through actions of Ca2+ calmodulin-dependent kinase II CaMKII) [63]. Thus, calcineurin effects in cardiomyocytes where mitochondrial membrane potential decreases caused by capsaicin could be through the disruption of the dephosphorylation/phosphorylation cycle of the TRPV1 channel. Furthermore, using a myocardial infarction rodent model, Hurt and collaborators demonstrated that capsaicin decreases the infarct area, an effect that was also blocked by capsazepine and by cyclosporine A suggesting that TRPV1 activation produces protective effects to the reperfusion injury in this myocardial infarction model  [51].”

Line 317-324. H/R reduced cell viability, which is which is potentiated by a high dose of capsaicin. This effect is achieved by increasing intracellular Ca2+ and triggering apoptosis in these cells [67]. Unexpectedly, H9C2 cells under H/R conditions also exhibited TRPV1 phosphorylation and capsaicin resulted in increased mitochondrial dysfunction, which was inhibited by capsazepine.

These sentences only described parts of fact. Hypoxia is known to induced hypoxia-inducible factor.

Hypoxia can directly damage mitochondria through the lack of oxygen to impede electron transfer through the electron transport chain.

Hypoxia upregulates HIF, which then upregulate TRPV1. I presume that the functions of mitochondria successfully influenced by the TRPV1 antagoinist, suggesting that TRPV1 may be the “effector” under hypoxia. Several papers support this idea. Please pain 2011 apr 152 (4):936-945 International scholar research notices 2011 Yu, C.S. : study on HIF-1a gene translation in psoriatic epidermis with the tpical treatment of capsaicin ointment.

Thanks for the comment the lines were rewritten and now it reads (lines: 290-310)

In contrast, other authors have reported that TRPV1 activation produces harmful effects for mitochondria and cardiac tissue. For instance, it was reported that H9C2 cells exposed to Hypoxia/Reoxygenation (H/R) conditions show reduced cell viability, which is potentiated by a high dose of capsaicin. This effect is achieved by increasing intracellular Ca2+, reducing the mitochondrial membrane potential and elevating mitochondrial superoxide production, leading to apoptosis of these cells [54]. Importantly, all of these effects are potentiated by capsaicin and attenuated by capsazepine. These evidences suggest that there are dose-dependent effects for capsaicin and that this compound promotes dual effects by either preventing or intensifying mitochondrial dysfunction in the H9C2 cells depending on the concentration of the vanilloid used in the experiments. Curiously, these cells display low TRPV1 expression, with channels mainly located at the ER where their activation increases cytosolic and mitochondrial Ca2+ levels, suggesting that TRPV1 channels localized at the ER could work as Ca2+ exchangers between the ER and the mitochondria [65], a role that requires to be studied with further detail. In addition, H9C2 cells exposed to H/R conditions display increases inTRPV1 phosphorylation, indicating channel sensitization. This finding is in agreement with a previous report performed using DRG neurons exposed to hypoxia conditions [66] where channel phosphorylation of serine residues increased due to the induction of the Hypoxia-inducible factor-1a (HIF-1α). The latter regulates protein kinase C (PKC)e translocation to the plasma membrane, then this kinase phosphorylates serine 800 located at the TRPV1 C-terminus [66]. This evidence reinforces that TRPV1 channel is sensitized by hypoxic conditions in different cell contexts, suggesting that capsaicin produces a synergic effect on a channel previously sensitized by an H/R condition in H9C2 cells resulting in cell death”.

Have you ever thought of a question: whether the phosphorylation of TRPV1, may not come from kinases, but from mitochondria? Therefore, the interaction between TRPV1 and mitochondria, like symbiotic , work together in against disease or causing disease? 

Thanks for the comment, until now there is no evidence that mitochondrial kinases phosphorylate TRPV1 channels. Furthermore, TRPV1’s topology in the mitochondrial membrane remains unknown although some evidences suggest that the intracellular domains could be at the intramitochondrial space. Hence, it is possible that mitochondrial kinases phosphorylate target residues located in these TRPV1 intracellular domains, but there are not experimental evidences supporting this. On the other hand, TRPV1 activation and mitochondria definitely function as symbiotic partners to produce or regulate disease depending on the cellular context and condition.

Additionally, we have modified the title and reorganized the sections in order to highlight that TRPV1 activation can regulate mitochondrial Ca2+ homeostasis independently of whether the channel is located in the cell surface or in intracellular membranes.

Round 2

Reviewer 2 Report

I'm satisfied to  see the authors quickly addressed all my questions. The review is not confusing in terms of clarifying mitochondria localized TRPV1 now. The written of English needs to be improved, there are some long sentence can be avoid. 

I cannot see Fig3 in the PDF.